# CA125 for the Diagnosis of Advanced Urothelial Carcinoma of the Bladder: A Systematic Review and Meta-Analysis

**DOI:** 10.3390/cancers15030813

**Published:** 2023-01-28

**Authors:** Hsuan-Jen Lin, Rouh-Mei Hu, Hung-Chih Chen, Chung-Chih Lin, Chi-Yu Lee, Che-Yi Chou

**Affiliations:** 1Division of Nephrology, Asia University Hospital, Taichung 41354, Taiwan; 2Department of Bioinformatics and Medical Engineering, Asia University, Taichung 41354, Taiwan; 3College of Medicine, China Medical University, Taichung 404332, Taiwan; 4Division of Nephrology and Kidney Institute, China Medical University Hospital, Taichung 404332, Taiwan; 5Department of Post-baccalaureate Veterinary Medicine, Asia University, Taichung 41354, Taiwan

**Keywords:** CA125, urothelial cancer, bladder cancer, diagnosis, meta-analysis

## Abstract

**Simple Summary:**

Urothelial carcinoma of the bladder (UCB) is a common genitourinary cancer. Advanced UCB, defined as muscle-invasive and lymph node-involving, is linked to poor outcomes. CA125 is a well-known tumor marker for ovarian cancer and can be used as a diagnostic marker for advanced UCB. This meta-analysis reviewed published studies in English and Chinese on serum CA125 for advanced UCB. One thousand five hundred seventy patients from 14 studies were analyzed. The diagnostic odds ratio was 8.138, and the area under the receiver operating characteristic curve was 0.797. Serum CA125 may help distinguish advanced UCB from non-advanced UCB patients. CA125 may help clinicians plan the treatment of UCB.

**Abstract:**

Background: Urothelial carcinoma of the bladder (UCB) is the second most common genitourinary cancer. This study aims to assess the diagnostic accuracy of CA125 in advanced UCB. Methods: We searched prevalent studies in PubMed, the Cochrane Library, Scopus, Embase, the Web of Science China National Knowledge Infrastructure database, and Wanfang data before October 2022. Pooled sensitivity, specificity, and summary receiver operating characteristics were used to assess the diagnostic value of CA125. Results: One thousand six hundred forty-one patients from 14 studies were analyzed. UCB stage T3–4N1 was defined as advanced UCB in ten studies; T2–4 was used in three studies; and N1M1 in one study. Patients’ age was between 21 to 92, and 21% to 48.6% of patients were female. The pooled sensitivity was 0.695 (95% confidence interval (CI): 0.426–0.875). The pooled specificity was 0.846 (95% CI: 0.713–0.924). The diagnostic odds ratio was 8.138 (95% CI: 4.559–14.526). The AUC was 0.797. Conclusion: CA125 may provide significant diagnostic accuracy in identifying muscle-invasive, lymph node-involved, and distant metastatic tumors in patients with urothelial carcinoma of the bladder. Limited studies have been conducted on the prognostic role of CA125. More studies are needed for a meta-analysis on the prognostic role of CA125 in UCB.

## 1. Introduction

Urothelial carcinoma of the bladder (UCB) is the 10th most common cancer, with approximately 573,000 new cases and 213,000 deaths worldwide. It is more common in men than in women. The incidence rate and mortality rate of UCB are four times higher in men than in women [1]. In the United States, an estimated 81,180 new cases of UCB were diagnosed, 61,700 were male and 19,480 were female. UCB is the 4th most common cancer in males. The estimated death was 17,100, including 12,120 males and 4980 females. UCB is the 8th most common cause of cancer death in the United States. Advanced age is the most significant risk factor for UCB, and the average age of diagnosis is between 70 and 84 years [2]. Smoking is an important modifiable exposure. The observed geographic patterns of UCB mirror the prevalence of smoking, with a population-attributable risk of approximately 50%. Occupational exposure to benzene dyes, factory chemicals, Schistosoma hematobium (particularly common in Northern Africa), cyclophosphamide, pelvic radiation, and arsenic contamination in drinking water also increases the risk of UCB [3]. In developing countries, the mortality rates of UCB have declined because of improvements in treatments such as endoscopic resection, adjuvant chemotherapy, and immunotherapy [4]. The incidence trends were stable in men from the 1990s to the 2020s, but the incidence trends were rising with the rising smoking prevalence in women [5]. As non-invasive cancers account for a large proportion of UCB, mortality rates may be of greater utility in assessing overall progress in controlling UCB [3].

The prognosis of UCB is strongly correlated with bladder muscle invasion. The 5-year survival is 93% in non-muscle-invasive, 36–48% in muscle-invasive, 36% in lymph nodes involved, and 5% in distant metastatic UCB [2]. Non-muscle-invasive UCB represents approximately 70% of organ-confined cancer and is treated differently from muscle-invasive UCB. Transurethral resection of the bladder tumor, a single perioperative dose of chemotherapy, and intravesical therapy are indicated in non-muscle-invasive UCB. Chemotherapy and radical cystectomy are indicated in muscle-invasive UCB. Adjuvant chemotherapy, cytotoxic chemotherapy, and immunotherapy, such as PD-L1, are indicated in metastatic UCB. Conventional staging modalities such as transurethral resection of bladder tumors (TURBT), computer tomography (CT), and magnetic resonance imaging (MRI) have a limited role in tumor staging and lymph node involvement in patients with muscle-invasive UCB. TURBT may diagnose UCB, resect the visible tumor, and allow for its staging. Repeat TURBT in 4–6 weeks is recommended in high-risk patients such as incomplete resection or pathological high-risk. 51% of patients may have residual disease, and 8% may have muscle-invasive UBC in the repeat TURBT [6]. MRI correctly staged 56% of patients, and CT correctly staged 50% of patients [7]. The Vesical Imaging-Reporting and Data System provides good diagnostic accuracy in identifying muscle-invasive UBC. The pooled sensitivity and specificity were 0.90 (95% CI: 0.86–0.94) and 0.86 (95% CI: 0.71–0.94) in a meta-analysis of more than 1000 patients from six studies [8]. However, the Vesical Imaging-Reporting and Data System is unable to identify the lymph nodes involved and metastatic UBC.

CA125 is a repeating peptide epitope of the mucin MUC16 and is involved in the glycosylation of MUC16 [9]. MUC16 overexpression promotes cancer cell proliferation [10] and cancer cell resistance to therapy [11]. CA125 is a well-known biomarker to monitor epithelial ovarian cancer and for the differential diagnosis of pelvic masses [12]. CA125 is routinely monitored in patients with ovarian cancer and is a prognostic indicator of cancer recurrence. CA125 may help identify advanced UCB because UCB is a type of epithelial cell, and MUC16 is overexpressed in advanced UCB [13]. CA125 expression in bladder epithelial cells is linked to the infiltration of immunosuppressive cells such as regulatory T cells and M2 macrophages. Clinical studies supported the applications of CA125, CA199, carcinoembryonic antigen (CEA) [14], vascular endothelial growth factors [15], and tumor-specific growth factor [16] in identifying advanced UCB. Monitoring the markers may help guide treatments such as TURBT or radical cystectomy. CA125 is the most extensively explored among these tumor markers in clinical studies. In this study, we defined muscle-invasive, lymph node-involved, and distant metastatic UCB as advanced UCB and non-muscle-invasive UCB as non-advanced UCB. In this meta-analysis, we assess the diagnostic accuracy of CA125 in advanced UCB among patients with UBC.

## 2. Methods

The systematic review followed the recommendations of the Preferred Reporting Items for Systematic Reviews and Meta-Analyses (PRISMA). The protocol has not been registered. We searched all published Chinese and English studies up to October 2022 that assessed the diagnostic values of serum CA125 for UCB in PubMed, Embase, Web of Science, Cochrane Library, Scopus, the China National Knowledge Infrastructure database (CNKI), and Wanfang data (Figure 1). The keywords were CA125 and bladder cancer or carcinoma, urothelial cancer or carcinoma, and transitional cell carcinoma. We selected articles using the following inclusion criteria: (1) patients with a pathological diagnosis of UCB, and the definition of advanced UCB was provided; (2) reporting the serum CA125 assay and cut-off; (3) reporting the number of patients according to the CA125 cut-off according to non-advanced and advanced UCB; (4) published in English or Chinese. Exclusion criteria were: (1) no serum CA125; (2) not advanced UCB; (3) studies using the same dataset; (4) case reports, letters, animal models, review articles, and meta-analyses; and (5) inadequate data, such as the number of the patients, according to the cut-off of CA125, was not available. We used the modified version of the quality assessment of diagnostic accuracy studies tool-2 (QUADAS-2) to assess the methodological quality of the selected studies.

Two researchers (HJ and HC) independently assessed the title, abstract, and full text to identify the relevant articles. Two researchers (CC and CY Lee) extracted the data. Disagreements were solved through discussions and by a third researcher (CY Chou).

## 3. Statistical Analysis

All analysis was performed using R Statistical Software (version 4.2.1, R Foundation for Statistical Computing, Vienna, Austria) with the mada and meta packages. A *p* < 0.05 was considered statistically significant. The true positive (TP), false positive (FP), false negative (FN), and true negative (TN) were extracted from the selected studies. The pooled sensitivity, specificity, diagnostic odds ratio, and summary receiver operating characteristics were obtained to assess the diagnostic value of CA125. The heterogeneity of studies was analyzed using the chi-square test. An I^2^ < 50% was considered as no observable heterogeneity, a fixed effects model was used, or a random effects model was utilized. The mada package did not provide a summary of sensitivity and specificity. We used the meta package to calculate the summary of sensitivity and specificity [17]. RevMan 5.2 software with QUADAS-2 was used to analyze the methodological quality.

## 4. Results

We identified 113 records from Pubmed, 22 from Embase, 70 from Scopus, 60 from Web of Science, 28 from CNKI, and 61 from WanFang data (Figure 1). The title and abstract of 242 records were screened after removing 311 duplicate records. The full text of 27 studies was obtained. Nine studies were excluded because the patient number, according to the cut-off for CA125, was unavailable. The data were extracted from 14 studies [14,18,19,20,21,22,23,24,25,26,27,28,29,30]. We summarized the cut-off of CA125, the patient number in each study, the age of the patients, and the percentage of female patients (Table 1). One thousand six hundred forty-one patients from 14 studies were analyzed. CA125 was measured using an enzyme immunoassay in nine studies [18,19,20,21,22,23,24,25,28]; five studies did not report the assay for CA125. A cut-off of 35 U/mL was used in 12 studies, 30 U/mL in one study, and 20 U/mL in another. UCB stage T3–4N1 was defined as advanced UCB in nine studies; T2–4 was used in two studies; and N1M1 was used in one study. Patients’ ages ranged from 21 to 92, and 10.9% to 49.1% were female.

The pooled sensitivity (Figure 2) was 0.695 [95% confidence interval (CI) 0.426–0.875] in the random effects model because the heterogeneity I^2^ = 91%. The pooled specificity was 0.846 (95% CI 0.713–0.924, I^2^ = 89%). The log diagnostic odds ratio (DOR, Figure 3) was 2.10 (95% CI: 1.52–2.68). The area under the receiver operating characteristic (AUC) was 0.797 (I^2^ = 22.4%) in the summary ROC analysis (Figure 4).

The methodological quality of 14 studies is summarized in Figure 5. One study [28] considered a high risk of bias, and four studies [21,25,29,30] considered an unclear risk of bias in patient selection. Three studies [25,26,30] had a high risk of bias in the index test. Four studies [22,25,26,30] were unclear regarding reference standards. Two studies [23,30] had a high risk of bias, and two [20,27] had an unclear risk of bias in the flow and timing. One study [28] had an applicability concern, and six studies [14,21,25,27,29,30] had an applicability concern in patient selection. In the index test, one study [30] had a high risk of applicability concerns, and three studies [23,25,26] had an unclear risk of applicability concerns. Five studies [21,22,25,26,30] had an unclear risk of applicability concerns in the reference standard.

## 5. Discussions and Systematic Review

### 5.1. CA125 for Advanced UCB

The diagnostic odds ratio of CA125 was 8.1381 (95% CI: 4.559–14.526), and the AUC was 0.797 for advanced UCB. Patients in the selected studies were around 70–80 years old, and the prevalence of males was four times higher than that of females. These characteristics of patients are consistent with that those of the national data from UCB [1,31]. A DOR of 8.138 and an AUC of 0.797 indicate an acceptable diagnostic accuracy in clinical practice. For example, C-reactive protein is wildly used in clinical practice to identify sepsis, and a DOR and AUC of C-reactive protein were 6.89 and 0.73, respectively [32]. The diagnostic accuracy can be further improved using a combination of biomarkers, such as CA199 and CEA [14,33]. We did not explore the accuracy of the combination of tumor markers in this study because the results were presented in different ways. [34,35] Margel et al. [14] reported the prognostic role of CA125 in advanced UCB. The combination of CA125, CA199, and CEA may improve the diagnostic accuracy for advanced UCB. Margel et al. developed and validated an algorithm [33,36] for patients at risk of advanced UCB.

Asymptomatic microscopic or gross hematuria is the most common presentation of UCB. Approximately 4% of patients with microscopic hematuria and 16.5% of those with gross hematuria have UCB [37]. Urine cytology is used in the evaluation of gross hematuria and posttreatment surveillance. The pooled sensitivity and specificity of the urine cytology were 0.37 (95% CI, 0.35–0.39) and 0.95 (95% CI, 0.94–0.95) in non-muscle-invasive UCB [38]. Urinary biomarkers are attractive in clinical setting because of their non-invasive nature. BTA, nuclear matrix protein 22, immunoCyt/uCyt+, UroVysion fluorescence in situ hybridization (FISH) assay are the US Food and Drug Administration-approved urinary biomarkers for non-muscle-invasive UCB [38]. However, none of those mentioned above biomarkers were explored for advanced UCB. We analyzed the diagnostic accuracy of CA125 in the diagnosis of UCB using three studies [20,22,39]. The pooled sensitivity was 0.776 (95% CI: 0.730–0.817, I^2^ = 0%), and the pooled specificity was 0.821 (95% CI: 0.577–0.939, I^2^ = 99%). The DOR was 18.018 (95% CI: 7.831–41.453), and the I^2^ was 51%. The AUC was 0.906 for CA125 for the diagnosis of UCB. Disease control (chronic kidney disease) patients were used as the reference group in our previous study [20] and may decrease the diagnostic accuracy of CA125.

The circulating tumor cells of UCB are linked to tumor stage, histological grade, metastasis, overall survival, progression/disease-free survival, and cancer-specific survival. The DOR of circulating tumor cells was 4.60 (2.34–9.03) for advanced UCB [40] and is lower than the DOR of CA125. The detection of plasma cell-free tumor DNA may predict a metastatic relapse after chemotherapy [41]. Compared to circulating tumor cells and plasma cell-free DNA, CA125 is available in most clinical settings without the need for assay validation and may help identify patients at risk of advanced UCB and direct the optimal treatment (cystoscopy or cystectomy).

### 5.2. CA125 as a Prognostic Marker

Four studies reported the association between CA125 and survival [18,24,29,34]. Elevated CA125 is linked to overall survival in advanced UCB but not recurrence-free survival in Chang et al.’s study [19]. However, CA125 was not related to survival in Ahmadi et al.’s work [18]. Ahmadi et al. showed that elevated CA199 or CEA is linked to overall and recurrence-free survival [18]. Margel et al. studied patients with organ-confined UCB after radical cystectomy. CA125 greater than 35 U/mL was associated with a higher chance of being lymph node-positive, extravesical [14], metastatic disease [29], and worse overall survival [24,29]. CA125 and CA199 were associated with worse overall survival and disease-specific survival in univariate Kaplan-Meier analysis. CA 199 was an independent predictor of disease-specific survival in multivariate Cox regression [34]. We calculated the pooled hazard ratio of CA125 and CA199 using data from two studies. [18,34] The pooled hazard ratio of CA125 for overall survival was 0.90 (95% CI 0.59–1.39, I^2^ = 0%). The pooled hazard ratio of CA199 for overall survival was 2.08 (95% CI 0.91–4.78, I^2^ = 66%). CA199 might provide a better prediction of survival than CA125. More studies are needed to confirm the role of CA199 in the prognosis of UCB.

Bazargani et al. reported that elevated CA125, CA199, or CEA (31% of 337 patients) were linked to worse recurrence-free survival (hazard ratio = 2.81, *p* < 0.001) and overall survival (hazard ratio = 3.97, *p* < 0.001) in Kaplan-Meier analysis [35]. In Izes et al.’s study, increased CA125 was also linked to therapeutic failure, and decreased CA125 reflected disease regression. The CA125 was 9.9 ± 5.1 U/mL in the patients without disease progression and 140.9 ± 155.4 U/mL in patients with radiologically documented progression [25]. Cook et al. explored tumor markers (beta-HCG, CEA, CA125, and CA199) in assessing chemotherapy response in advanced UCB. Clinical response was strongly related to marker response [42]. Manvar et al. reviewed studies using a different cut-off for CA125 (15, 20, and 35 U/mL). A 15 or 20 U/mL threshold resulted in a higher proportion of upstaged UCB [43]. A lower cut-off of CA125 improves sensitivity for predicting advanced UCB at the expense of increasing the false-positive rate in patients with non-advanced UCB [28].

### 5.3. Correlation of CA125 with Tumor Stage and Grade

Ten of the 14 studies [18] reported a significant association between CA125 and tumor stage. Patients with a CA125 greater than 35 U/mL had a higher chance of developing muscle-invasive, lymph node-involved, and metastatic disease. [14,19,21,22,23,24,27,28,29]. The percentage of patients with a CA125 greater than 35 U/mL was 6–11% in T1–T2 UCB and 36–92% in T3–4 UCB. Three studies reported that CA125 was significantly higher in T3–4, lymph node-involved disease, and metastatic disease than in the non-advanced UCB [24,27,28,29]. Margel et al.’s study also supported the association between CA125 and tumor stage because the ratio of CA125 greater than 35 U/mL was significantly higher in extravesical disease and lymph node-involved disease [14]. No association between CA125 and tumor stage was found in Ahmadi et al.’s and Bazargani et al.’s works [18,35]. Patients with CA125 greater than 35 U/mL had a higher chance of having high-grade UCB [22,23,24]. CA125 was significantly higher in patients with high-grade UCB [21,24,27]. Higher grade UCB is linked to resistance to treatment and recurrence. Three studies reported the histological grade as G1-3 [22,26,27]. The percentage of patients with a CA125 greater than 35 U/mL was 5.9–26.3% in G1, 12.5–68.4% in G2, and 72–92% in G3.

### 5.4. CA125 for Metastatic UCB

Four studies supported the association between elevated CA125 and metastatic UCB [24,25,26,29]. Three studies (110 patients) reported a high number of patients with a CA125 greater than 35 U/mL and metastatic UCB [24,25,26,29]. The pooled sensitivity was 0.850 (95% CI 0.624–0.951, I^2^ = 0%), and the pooled specificity was 0.697 (95% CI: 0.553–0.811, I^2^ = 54%). The DOR was 8.516 (95% CI: 2.403–30.186), and the I^2^ was 0%. The AUC was 0.803 for CA125 in the diagnosis of metastatic UCB. CA125 was significantly higher in patients with metastatic UCB, with an average CA125 of 26.4 ± 6.4 U/mL (20.4 ± 7.0 in non-metastatic UCB) [24]. There were six patients with metastatic UCB in the Yang et al. study, and 100% of them had CA125 of more than 35 U/mL [23].

### 5.5. CA125 in Follow-Up

Three [23,25,27] of the fourteen studies explored changes in serum CA125 after treatment and recurrence. In Izes et al.’s study, CA125 was reduced by 42% after chemotherapy. 70% of patients with increased CA125 had radiologically documented progression. Five cases with increased CA125 had clinical deterioration during the next several months [25]. The average CA125 of 15 patients who underwent urinary diversion operations decreased from 398 U/mL to 368 U/mL after the surgery. The serum CA125 of 10 patients who underwent TURBT was reduced from 480 U/mL to 368 U/mL. The serum CA125 of 17 patients decreased from 475 to 237 U/mL after embolization. CA125 was increased in 16 patients with recurrence in a 3–12 month follow-up [27]. In Yang et al.’s study, 54.3% of patients had a recurrence in a 3–24 month follow-up. 63.7% of patients had a serum CA125 greater than 35 U/mL (compared with 28.5% in patients without recurrence) [23]. In Bazargani et al.’s study, 51% (30/59) of patients had elevated CA125, CA199, or CEA. 10% of patients with normal levels of CA125, CA199, and CEA had a lower rate and a longer median time to recurrence/progression [35].

We use QUADAS-2 to assess the methodological quality. In the patient selection bias, patients were enrolled for radical cystectomy, or TURBT, in the selected studies. No study reported other inclusion criteria. Because the serum CA125 was taken before the surgery in all patients, we did not consider the enrollment of the UCB patients as a patient selection bias. In the reference standard, two studies selected 30 and 20 U/mL as the optimal cut-off because the cut-off was not set based on the AUC analysis. We assigned an unclear risk of reference standard bias to the studies.

Some limitations may be considered when interpreting the conclusion of the study. Six [21,22,23,24,26,29] of the fourteen studies were published in Chinese, and an English abstract was available for four [21,22,24,26] of the six studies. We checked the quality of all studies, and these studies were suitable for data extraction. Because half of the included patients may be Chinese, the applicability of the results to different ethnicities and regions may be reduced. The incidence of UCB is highest in southern Europe (Greece (highest incidence rate in men), Spain, and Italy), followed by Western Europe (Belgium and the Netherlands), and Northern America. The highest rate is in Hungary among women. No reports are available from these countries or regions. The studies from China showed consistent results on the diagnostic accuracy of CA125 in advanced UCB. The studies that showed different results were from the United States. More studies are needed to explore if the diagnostic accuracy of CA125 is related to the incidence of UCB in different countries or regions. The meta-analysis was performed cross-sectionally, which may lead to an overestimation of the performance of serum CA125. A significant degree of performance heterogeneity was noted and may affect the accuracy of serum CA-125 in the diagnosis of UCB. Although we did not detect substantial publication bias, we cannot completely exclude the possibility of publication bias. Finally, we searched all the available databases for eligible studies. However, we might not include all studies on CA125 and UCB.

## 6. Conclusions

CA125 may provide significant diagnostic accuracy in identifying muscle-invasive, lymph node-involved, and distant metastatic tumors in patients with urothelial carcinoma of the bladder. Significant heterogenous results were found in the selected studies with regard to the diagnostic accuracy of CA125 in identifying advanced from non-advanced urothelial carcinoma of the bladder. A limited number of studies on the prognostic role of CA125 were available. More studies are needed to conduct a meta-analysis on the prognostic role of CA125 in UCB.

## Figures and Tables

**Figure 1 cancers-15-00813-f001:**
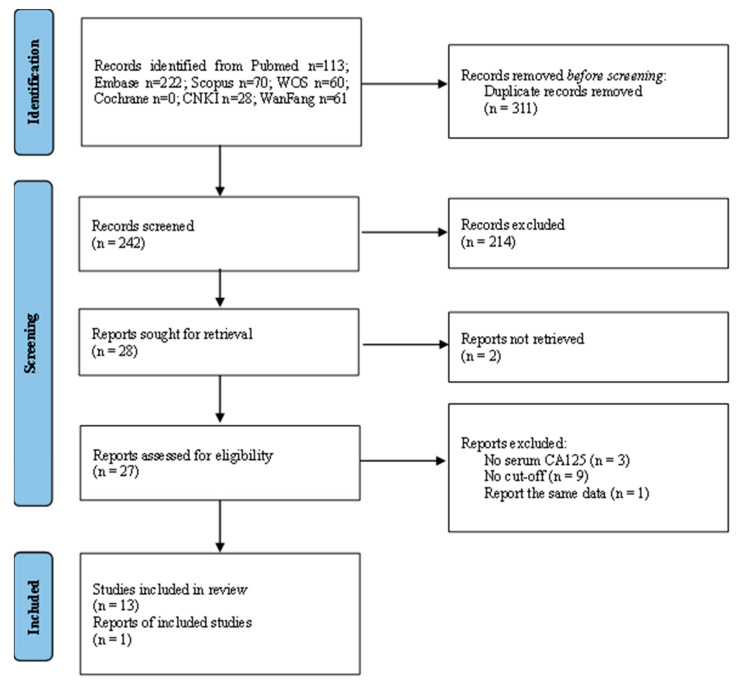
Flow diagram for systematic review. WOS: web of science; CNKI: China National Knowledge Infrastructure.

**Figure 2 cancers-15-00813-f002:**
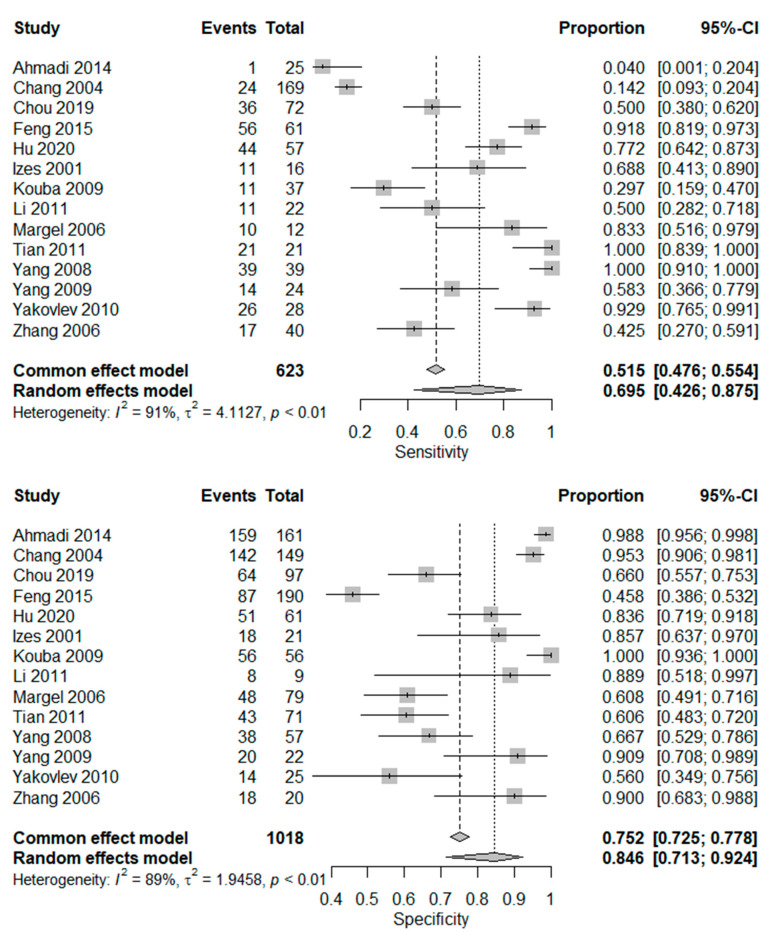
Forest plots of the pooled sensitivity and specificity of CA125 for advanced urothelial carcinoma of the bladder [14,18,19,20,21,22,23,24,25,26,27,28,29,30].

**Figure 3 cancers-15-00813-f003:**
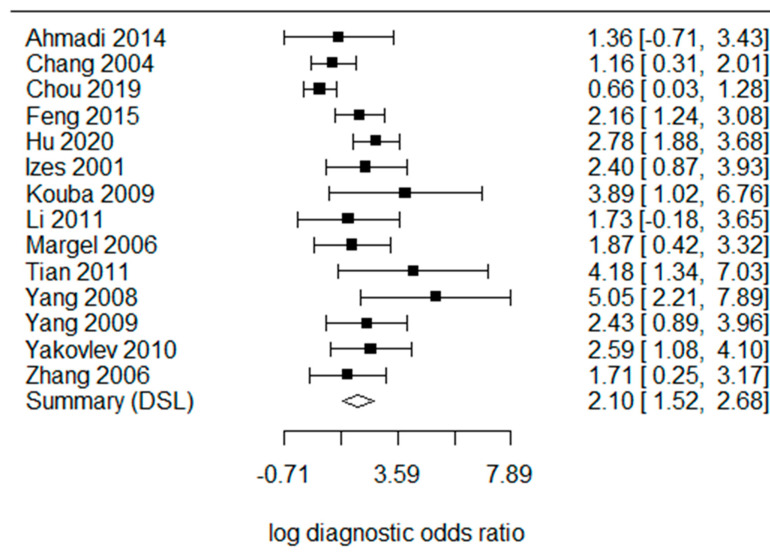
Forest plots of the diagnostic odds ratio of CA125 in the diagnosis of advanced urothelial carcinoma of the bladder [14,18,19,20,21,22,23,24,25,26,27,28,29,30].

**Figure 4 cancers-15-00813-f004:**
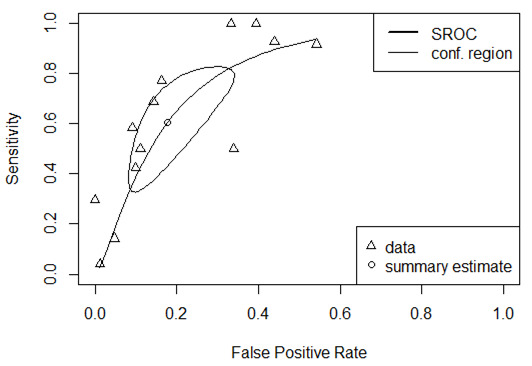
The summary receiver operating characteristic (SROC) curve of CA125 in the diagnosis of advanced urothelial carcinoma of the bladder.

**Figure 5 cancers-15-00813-f005:**
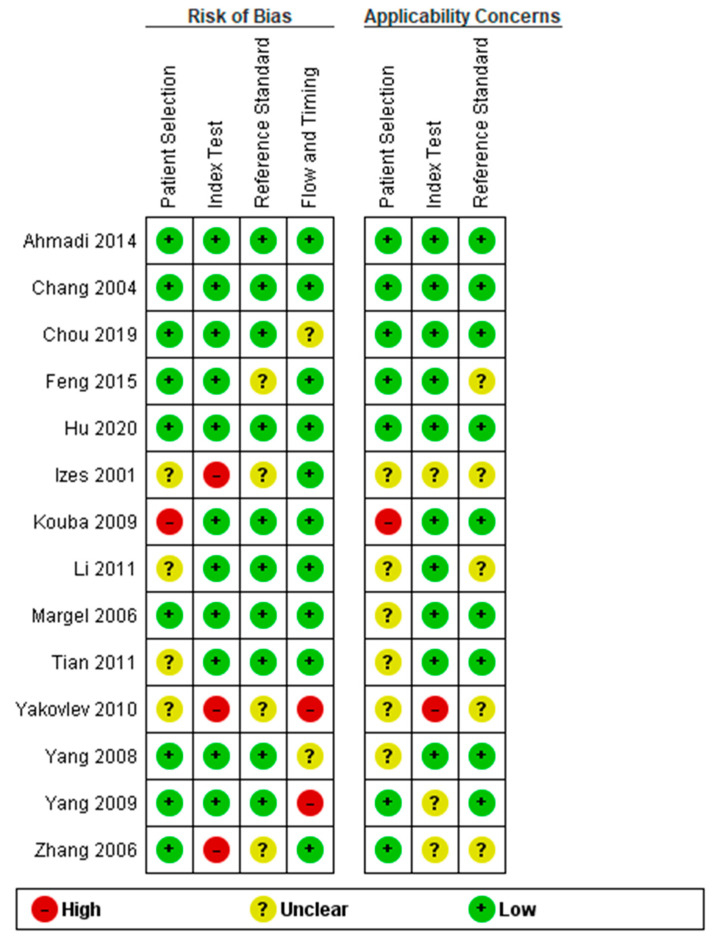
Summary of methodological quality of all studies using the QUADAS-2 [14,18,19,20,21,22,23,24,25,26,27,28,29,30].

**Table 1 cancers-15-00813-t001:** Characteristics of included studies reporting CA125 for advanced urothelial carcinoma of the bladder (UCB) [14,18,19,20,21,22,23,24,25,26,27,28,29,30].

Study	Cut-Off	Advanced	Number	Age	Female (%)
Ahmadi 2014	35	T3-4N1	186	36–89	36.6
Chang 2004	35	T3-4N1	287	34–89	21.3
Chou 2019	35	T3-4N1	169	37–90	35.5
Feng 2015	35	T3-4N1	251	21–92	48.6
Hu 2020	35	T3-4N1	118	55–79	49.1
Izes 2001	35	T3-4N1M1	68	43–86	NA
Kouba 2009	35	T3-4N1	92	NA	31.5
Li 2011	35	T2-4	90	31–88	26.7
Margel 2006	35	T3-4N1	91	67 ± 9	15.4
Tian 2011	35	T3-4N1	92	47–85	40.2
Yang 2008	35	T3-4M1	58	45–86	22.4
Yang 2009	35	T2-4	46	43–74	10.9
Yakovlev 2010	30	N1M1	53	45–71	24.5
Zhang 2006	20	T2-4	60	41–75	30.0

NA: Not available.

## Data Availability

The data used to support this study are available from the corresponding author upon request.

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
