# Peer review of "CA125 for the Diagnosis of Advanced Urothelial Carcinoma of the Bladder: A Systematic Review and Meta-Analysis"

_cancers, 2023, doi:10.3390/cancers15030813_

Round 1

Reviewer 1 Report

The authors deal with a still interesting issue in oncology, the contemporary role of biomarkers in bladder cancer.

This well written paper, as interesting as it is, is compromised by some flaws.

Background

The authors describe in a long winded way general considerations regarding the bladder cancer (BC) which is superfluous. (line 36 – 85). The briefing of CA125 is 4 lines long.

The authors should focus on BC and tumor markers in general and CA125 in this specific case. They should explain why CA125 is a useful marker. In reality CA125 plays a subordinated role and only in few centers CA125 is used.

Methods

Nothing to comment

Discussion

The statement in line 171 is wrong. The authors did not show the usefulness of CA125 in the diagnosis of BC.

They should follow a certain systematic:

1.    The value of CA125 as a diagnostic marker

2.    The value as a prognostic marker

3.    The correlation of CA125 with tumor stages

4.    The value of CA125 in metastatic disease

5.    The value of CA125 in the follow up

Conclusions

They are bold and do not coincide with the real life. More objectivity is required.

Author Response

The authors describe in a long winded way general considerations regarding the bladder cancer (BC) which is superfluous. (line 36 – 85). The briefing of CA125 is 4 lines long.

Response to the comment: Thank you. We shortened the introduction of UCB and added more background on CA125. 

The authors should focus on BC and tumor markers in general and CA125 in this specific case. They should explain why CA125 is a useful marker. In reality CA125 plays a subordinated role and only in few centers CA125 is used.

Respond to the comment: We agreed with the reviewer that CA125 is used in a few centers. The study may provide some evidence to support the clinical application of CA125 in identifying advanced UCB.

Discussion

The statement in line 171 is wrong. The authors did not show the usefulness of CA125 in the diagnosis of BC.

They should follow a certain systematic:

1.    The value of CA125 as a diagnostic marker

2.    The value as a prognostic marker

3.    The correlation of CA125 with tumor stages

4.    The value of CA125 in metastatic disease

5.    The value of CA125 in the follow up

Respond to the comments: Thank you for the comment. We re-organized the systematic review as suggested.

Conclusions

They are bold and do not coincide with the real life. More objectivity is required.

Response to the comment: Thank you. We rewrite the conclusions as you suggested.

Reviewer 2 Report

The introduction is quite elaborate and mentions etiology , staging which are supposed to be known to the reader already I suggest that introduction needs to be cut short and limited to the introduction of topic that is being studied, as in this article it is about CA125. . The authors can omit other details about TCC Bladder 

Author Response

The introduction is quite elaborate and mentions etiology , staging which are supposed to be known to the reader already I suggest that introduction needs to be cut short and limited to the introduction of topic that is being studied, as in this article it is about CA125. . The authors can omit other details about TCC Bladder.

Response to the comment: Thank you for the comment. We shortened the introduction as suggested.

Round 2

Reviewer 1 Report

The manuscript is improved. But the focus on CA125 in the bladder cancers in the introduction section is still missing . All the epidemiological data can be merged. The relation between CA125 and BC is the only relevant and interesting issue.

Author Response

The manuscript is improved. But the focus on CA125 in the bladder cancers in the introduction section is still missing. All the epidemiological data can be merged. The relation between CA125 and BC is the only relevant and interesting issue.

Respond to the comment: Thank you for the comment. We added the association of CA125 and advanced UCB in page 2 line 76-91.  We merged the epidemiological data as suggested.